# Relationship between Diabetic Nephropathy and Development of Diabetic Macular Edema in Addition to Diabetic Retinopathy

**DOI:** 10.3390/biomedicines11051502

**Published:** 2023-05-22

**Authors:** Yukihisa Suzuki, Motohiro Kiyosawa

**Affiliations:** 1Department of Ophthalmology, Japan Community Health Care Organization, Mishima General Hospital, Shizuoka 411-0801, Japan; 2Research Team for Neuroimaging, Tokyo Metropolitan Institute of Gerontology, Tokyo 173-0015, Japan; 3Jiyugaoka Kiyosawa Eye Clinic, Tokyo 152-0035, Japan; nra12337@nifty.com

**Keywords:** diabetic macular edema, diabetic nephropathy, diabetic retinopathy, risk factors

## Abstract

This study aimed to examine the relationship between diabetic retinopathy (DR) and systemic factors. We evaluated 261 patients (143 men, 118 women, aged 70.1 ± 10.1 years) with type 2 diabetes. All participants underwent a fundus examination, fundus photography using spectral domain optical coherence tomography (SD-OCT), and blood tests. For glycated hemoglobin (HbA1c) levels, the average and highest values in the past were used. We observed DR in 127 (70 men and 57 women) of 261 patients. Logistic regression analyses revealed a significant correlation between DR development and the duration of diabetes (OR = 2.40; 95% CI: 1.50), average HbA1c level (OR = 5.57; 95% CI: 1.27, 24.4), highest HbA1c level (OR = 2.46; 95% CI: 1.12, 5.38), and grade of diabetic nephropathy (DN) (OR = 6.23; 95% CI: 2.70, 14.4). Regression analyses revealed a significant correlation between the severity of DR and duration of diabetes (*t* = –6.66; 95% CI: 0.21, 0.39), average HbA1c level (*t* = 2.59; 95% CI: 0.14, 1.02), and severity of DN (*t* = 6.10; 95% CI: 0.49, 0.97). Logistic regression analyses revealed a significant correlation between diabetic macular edema (DME) development and DN grade (OR = 2.22; 95% CI: 1.33, 3.69). DN grade correlates with the development of DR and DME, and decreased renal function predicts the onset of DR.

## 1. Introduction

Diabetes is a disease that causes chronic hyperglycemia due to insufficient insulin secretion or insulin action. The number of people with diabetes worldwide was estimated at 420 million in 2015. This number is currently on the rise and is expected to reach 640 million by 2040 [1]. There are two main types of diabetes: insulin-dependent type 1 diabetes and non-insulin-dependent type 2 diabetes. In type 1 diabetes, almost no endogenous insulin is secreted due to the dysfunction of the insulin-producing beta cells. Type 2 diabetes is a disease in which endogenous insulin is secreted to some extent, but hyperglycemia occurs due to impaired insulin secretion or insulin resistance [2]. Complications of diabetes include retinopathy, nephropathy, and neuropathy, which can occur in both type 1 and type 2 diabetes.

Diabetic retinopathy (DR) is a microvascular complication caused by persistent hyperglycemia that is common in diabetic patients and is a major cause of blindness along with glaucoma and age-related macular degeneration [3]. DR is classified into non-proliferative DR and proliferative DR, according to the degree of disease progression, and may be complicated by macular edema. Patients with nonproliferative DR are typically asymptomatic. If proliferative DR develops, the patient may present with a sudden loss of vision due to a vitreous hemorrhage. In type 2 diabetes, the incidence with 5 years of evolution is 20%, while with 15 years of evolution, it reaches 80% [4]. The risk of developing DR is thought to be associated with the duration of diabetes, hypertension, hyperlipidemia, and glycated hemoglobin (HbA1c) levels [5]. HbA1c levels have been reported to be significantly associated with the progression of DR [6], and glycemic variability has been found to be associated with DR in type 2 diabetes [7]. In many patients with diabetes, HbA1c levels fluctuate from month to month and are thought to affect the retina over the next several decades. We thought it necessary to investigate the effects of the average HbA1c level and highest HbA1c level over the past years on the development and severity of retinopathy, respectively.

Macular edema (ME) can develop in various diseases such as uveitis [8], retinal vein occlusion [9], and diabetic macular edema [10] and is a major cause of vision loss. In macular edema, fluid leaking from capillaries accumulates in the subretinal or intraretinal region of the macula [11]. Diabetic macular edema (DME) also occurs with some frequency in non-proliferative DR and is a major cause of vision loss in such patients [12]. DME is defined as “retinal thickening within one disk diameter of the center of the macula or definite hard exudates in this region” [13]. The risk factors for developing DME include long-term diabetes, hypertension, and high HbA1c levels [14].

Diabetic nephropathy (DN) is also a major and important microvascular complication in diabetes [15]. DN is the leading cause of chronic kidney disease and accounts for 40% of new cases of end-stage renal disease each year. DN is characterized by persistent albuminuria and decreased glomerular filtration rate (GFR) and also causes elevated blood pressure. In patients with DN, persistent albuminuria reflects glomerular injury but may also reflect generalized endothelial dysfunction and widespread vascular injury [16,17]. Several studies have reported that DN is associated with the development and progression of DR [6,18]. DN generally causes body edema along with hypertension and proteinuria [19]. In patients with DN, hyperpermeability of the retinal blood vessels is expected to occur due to microvascular abnormalities, which are presumed to cause an increased leakage of serum into the extracellular space [20].

We hypothesized that DR development and severity are related not only to past average HbA1c levels but also to the highest HbA1c level. We also hypothesized that DME development is associated with DN. We investigated risk factors for developing DR and DME in patients with type 2 diabetes using logistic regression analyses.

## 2. Materials and Methods

### 2.1. Standard Protocol Approvals, Registrations, and Patient Consents

The study protocol was approved by the institutional ethics committee of the Japan Community Health Care Organization, Mishima General Hospital (protocol code: H30-005 and date of approval: 5 October 2018). All the procedures conformed to the tenets of the Declaration of Helsinki. All measurements in this study were performed at the Japan Community Health Care Organization Mishima General Hospital. Informed consent was obtained from all participants prior to participation in the study.

### 2.2. Subjects and Examination

This study is retrospective consecutive case series. Our study included 261 patients (143 men, 118 women, aged 70.1 ± 10.1 years) with type 2 diabetes who visited the Department of Ophthalmology, Mishima General Hospital. Type 1 diabetes and type 2 diabetes are inherently different pathologies [1,2]. It has been also reported that the incidence of DR differs between type 1 diabetes and type 2 diabetes [21,22], and in this study, only type 2 diabetes was focused on. Individuals with diabetes were classified as those with fasting plasma glucose >126 mg/dL or casual blood glucose >200 mg/dL and a previous diagnosis of diabetes by an internist. Inclusion criteria included patients who had regular medical examinations and required blood tests (HbA1c, creatinine) and urinalysis (urinary protein, urinary albumin). Patients with HbA1c data at least once every 6 months and at least the past 7 years were included (Figure 1). Exclusion criteria were lack of HbA1c data for the past 7 years and being unable to perform urinalysis due to chronic renal failure. We performed fundus examination and fundus photography using spectral-domain optical coherence tomography (SD-OCT) after mydriasis in all cases. OCT scans were acquired using an RS-3000 device (NIDEK, Gamagori, Japan), capturing an area of 9 × 9 mm^2^ centered on the fovea. We performed fluorescein angiography to determine the grade of DR in patients with DR. In addition, the presence or absence of DME was determined from SD-OCT images. Blood tests (HbA1c, creatinine, LDL, triglyceride) and urinalysis (urinary protein, urinary albumin) were taken from ophthalmological examinations within 2 months, if available, or newly performed. The estimated glomerular filtration rate (eGFR) in each case was calculated from the blood creatinine level and age. Based on these data, we examined the stages of DR and DN in each case. The new Fukuda classification was used to diabetic retinopathy [23]. DR is divided into benign (type A) and malignant (type B), each of which is divided into five stages. Benign retinopathies include background retinopathy (A1 and A2) before proliferative changes and interrupted proliferative retinopathy (A3, A4, and A5) after photocoagulation or vitrectomy. Benign retinopathy includes background retinopathy (A1 and A2) and interrupted proliferative retinopathy (A3, A4, and A5) after photocoagulation or vitrectomy. DME is defined as a retinal thickening within 500 μm of the fovea or hard exudates at or within 500 μm of the fovea associated with adjacent retinal thickening or an area of retinal thickening that is one disc area or larger in size located one disc diameter (1500 μm) or less from the fovea [24]. We defined DME severity as none, mild (identified by retinal thickening or hard exudates in the posterior pole but distant from the center of the macula), moderate (characterized by retinal thickening or hard exudates approaching the center of the macula but not affected by the center), and severe (characterized by retinal thickening or hard exudates affecting the center of the macular) [25]. The diabetic nephropathy staging [26] was used for DN staging. The diabetic nephropathy staging classifies nephropathy into five stages from the 1st stage (early stage of nephropathy) to the 5th stage (dialysis therapy stage) using urinary albumin level or urinary protein level and eGFR. Urinary albumin levels were classified into A1 (less than 30 mg/day), A2 (30–299 mg/day), and A3 (300 mg/day or more) according to the KINGO CKD guideline 2012. We examined each patient for a history of ischemic heart disease and use of rennin-angiotensin-aldosterone system (RAS) inhibitors and sodium/glucose cotransporter (SGLT)-2 inhibitors.

After resting for 2 min, the blood pressure of each patient was measured in a sitting position at our hospital. Systemic hypertension was defined as systolic blood pressure of 140 mmHg or higher or diastolic blood pressure of 90 mmHg or higher.

We collected data on blood tests (HbA1c, creatinine, LDL, triglyceride) and urinalysis (urinary protein, urinary albumin), fundus examination, and fundus photography using spectral-domain optical coherence tomography, and blood pressure. HbA1c data were collected at least once every 6 months and for at least the past 7 years.

### 2.3. Data Processing and Analysis of Risk Factors for Development and Severity of DR

The DR and DN stages in each case were converted into numerical values (retinopathy, 0–7, nephropathy, 1–5). We performed multivariable logistic regression analyses on all patients with type 2 diabetes to investigate the risk factors for developing DR. We selected the eye with more severe DR as the target eye for each patient. Cases with DR developing in only one eye were included in patients with DR. We used the Kolmogorov–Smirnov test to examine whether DR stage, average HbA1c level, high HbA1c level, and eGFR were normally distributed.

Multivariable logistic regression analyses were performed to determine the association between the parameters (gender, duration of diabetes, body mass index, systemic hypertension, average HbA1c level, highest HbA1c level, serum LDL, and serum triglyceride, DN stage, ischemic heart disease, and use of RAS inhibitors and SGLT-2 inhibitors) and a diagnosis of type 2 diabetes. Odds ratios (OR) and 95% confidence intervals (CI) were calculated for each factor. Statistical significance for all analyses was defined as *p* < 0.05.

We also performed a regression analysis between DR stage and the parameters (gender, duration of diabetes, body mass index, systemic hypertension, average HbA1c level, highest HbA1c level, serum LDL, and serum triglyceride, DN stage, ischemic heart disease, and use of RAS inhibitors and SGLT-2 inhibitors) in patients with DR. If the DR stages of the left and right eyes were different, the more severe stage was considered as the DR stage of the patient. The *t*-statistic and 95% CI were estimated for each factor. Statistical significance for all analyses was defined as *p* < 0.05.

These regression models were constructed by identifying potential confounding variables. All analyses were conducted using EZR [27] (Saitama Medical Center, Jichi Medical University, Saitama, Japan), a graphical user interface for R (The R Foundation for Statistical Computing, Vienna, Austria).

### 2.4. Analysis of Risk Factors for Developing DME

We performed multivariable logistic regression analyses on all patients with type 2 diabetes to investigate the risk factors for DME. Multivariable logistic regression analyses were performed to determine the association between the parameters (gender, duration of diabetes, body mass index, systemic hypertension, average HbA1c level, highest HbA1c level, serum LDL, serum triglyceride, DN stage, ischemic heart disease, and use of RAS inhibitors and SGLT-2 inhibitors) and the presence of DME. Cases with DME developing in only one eye were included in patients with DME. The *t*-statistic and 95% CI were estimated for each factor. Statistical significance for all analyses was defined as *p* < 0.05.

We also performed a regression analysis between DME stage and the parameters (gender, duration of diabetes, body mass index, systemic hypertension, average HbA1c level, highest HbA1c level, serum LDL, and serum triglyceride, DN stage, ischemic heart disease, and use of RAS inhibitors and SGLT-2 inhibitors) in patients. If the DME stages of the left and right eyes were different, the more severe stage was considered the DME stage of the patient. The *t*-statistic and 95% CI were estimated for each factor. Statistical significance for all analyses was defined as *p* < 0.05.

### 2.5. Data Availability

All data used for analysis are presented in the tables in this article. Data will be provided in anonymized form after ethics approval if requested by other investigators for purposes of replicating the results.

## 3. Results

### 3.1. Our Patients with Type 2 Diabetes

We observed 127 (70 men and 57 women) patients with DR and 134 (73 men and 61 women) without DR (Table 1). The diabetes duration was significantly longer in patients with DR than in patients without DR. The average and highest HbA1c levels were significantly higher in patients with DR than in patients without DR. In addition, eGFR was low in patients with DR, and the DN grade was also high in patients with DR. On the other hand, DME was observed in 64 (50.4%) patients with DR but not in patients without DR. There were no differences in body mass index, prevalence of systemic hypertension, serum LDL, and serum triglyceride.

### 3.2. Kolmogorov-Smirnov Test

The DR grade (*p* < 0.001), average HBA1c level (*p* = 0.001), and high HBA1c level (*p* = 0.002) were not normally distributed, while eGFR was normally distributed.

### 3.3. Risk Factors for the Development of DR

Multivariate logistic regression analyses revealed a significant correlation between DR development and the diabetes duration (OR = 2.77; 95% CI: 1.14, 2.39; *p* = 0.009), average HbA1c level (OR = 5.60; 95% CI: 1.36, 23.1; *p* = 0.02), highest HbA1c level (OR = 2.46; 95% CI: 1.12–5.38; *p* = 0.02), and DN stage (OR = 7.62; 95% CI: 2.63, 22.1; *p* = 0.0002) (Table 2). Univariate analysis revealed a significant correlation between DR development and the eGFR (OR = 0.98; 95% CI: 0.97, 0.99; *p* = 0.0001), albuminuria (OR = 5.23; 95% CI: 3.14, 8.71; *p* < 0.00001) (Table 3). No significant correlation was observed between DR development and the other parameters.

### 3.4. Risk Factors for Severity of DR

Multivariate regression analyses revealed a significant correlation between DR severity and diabetes duration (*t* = 4.97; 95% CI: 0.15, 0.36; *p* < 0.00001), average HbA1c level (*t* = 3.25; 95% CI: 0.14, 1.02; *p* = 0.002), and DN stage (*t* = 4.32; 95% CI: 0.33, 0.89; *p* = 0.00004) (Table 4). Univariate analysis revealed a significant correlation between DR severity and the eGFR (OR = –4.96; 95% CI: –0.04, –0.020; *p* < 0.00001), albuminuria (OR = 0.19; 95% CI: 1.24, 1.99; *p* < 0.00001) (Table 5). No significant correlation was observed between the DR stage and the other parameters.

### 3.5. Risk Factors for the Development of DME

Multivariate logistic regression analyses revealed a significant correlation between DME development and the diabetes duration (OR = 1.33; 95% CI: 1.01, 1.75; *p* = 0.04), DN stage (OR = 2.80; 95% CI: 1.37, 5.72; *p* = 0.005) (Table 6). Univariate analysis revealed a significant correlation between DME development and the eGFR (OR = 0.98; 95% CI: 0.97, 0.99; *p* = 0.009), albuminuria (OR = 4.05; 95% CI: 2.30, 7.11; *p* < 0.00001) (Table 7). No significant correlation was observed between DME development and the other parameters.

### 3.6. Risk Factors for Severity of DME

Multivariate regression analyses revealed a significant correlation between DME severity and diabetes duration (*t* = 2.57; 95% CI: 0.02, 0.13; *p* = 0.001), average HbA1c level (*t* = 2.54; 95% CI: 0.08, 0.68; *p* = 0.01), and DN stage (*t* = 2.48; 95% CI: 0.04, 0.36; *p* = 0.002) (Table 8). Univariate analysis revealed a significant correlation between DR severity and the eGFR (OR = –2.44; 95% CI: –0.009, –0.001; *p* = 0.02), albuminuria (OR = 5.36; 95% CI: 0.26, 0.55; *p* < 0.00001) (Table 9). No significant correlation was observed between the DRE stage and the other parameters.

### 3.7. DME Prevalence by DR Grade

Our patients included 134 patients with Fukuda classification A0, 3 patients with A1, 49 patients with A2, 34 patients with B1, 7 patients with B2, 23 patients with B4, and 3 patients with B5, respectively (Table 10). DME prevalence was 32.7% in A2, 58.8% in B1, 71.4% in B2, 87.0% in B4, and 100% in B5, respectively. Additionally, DME was not observed in A0 and A1.

## 4. Discussion

We observed 127 (70 men and 57 women) patients with DR and 134 (73 men and 61 women) without DR. The diabetes duration was longer in patients with DR than in patients without DR. The average and highest HbA1c levels were higher in patients with DR than in patients without DR. Multivariate logistic regression analyses revealed a significant correlation between DR development and the duration of diabetes (OR = 1.65), average HbA1c level (OR = 5.60), highest HbA1c level (OR = 1.17), and DN grade (OR = 7.62). Multivariate logistic regression analyses revealed a significant correlation between DR stage and diabetes duration, average HbA1c level, and DN stage. We observed a significant correlation between DME development and diabetes duration (OR = 1.33), average HbA1c level (OR = 5.52), and DN grade (OR = 2.80). Multivariate regression analyses revealed a significant correlation between DME stage and diabetes duration, average HbA1c level, and DN stage.

### 4.1. Pathophysiology of DR

The primary retinal vascular response to exposure to hyperglycemia is vasodilation and changes in blood flow. These responses are thought to be metabolic autoregulation to increase retinal metabolism in diabetic patients [28]. On the other hand, high glucose triggers apoptosis of capillary pericytes, resulting in the localized outpouching of the capillary walls. This process results in the formation of a microaneurysm, which is the earliest clinical manifestation of DR [29]. In parallel with this, endothelial cell apoptosis and basement membrane thickening in diabetic microvessels, which are associated with blood-retinal barrier (BRB) impairment, have been reported [30]. Furthermore, the loss of vascular pericytes and endothelial cells induces capillary occlusion and causes retinal ischemia. Retinal ischemia and hypoxia activate hypoxia-inducible factor 1 (HIF-1), leading to increased VEGF expression in the eye [31]. VEGF is a growth factor that induces angiogenesis and angiogenesis by stimulating the division, migration, and differentiation of vascular endothelial cells, and it is associated with the progression of PDR and DME. In addition, VEGF has a vascular hyperpermeability effect [32,33]. Adamis et al. [34] observed that VEGF was increased in the vitreous of patients with PDR. In the present study, the development of DR was associated with both the average and highest HbA1c levels, and the severity of DR was associated with the average HbA1c level. Song et al. [6] conducted a 3-year retrospective cohort study of 604 patients with type 2 diabetes. They observed that the mean HbA1c level was a significant predictor of DR progression, independent of the duration of diabetes and HbA1c-variability levels. In patients with type 2 diabetes mellitus, HbA1c variability such as microalbuminuria and decreased GFR is associated with DN initiation [35]. In recent studies, glycemic variability has been found to be strongly associated with DR in patients with type 2 diabetes [7]. It is unclear why HbA1c variations adversely affect the development of DR or DN, but one possible mechanism is associated with “metabolic memory” due to repeated exposure to glycemic instability [36], which can lead to increased oxidative stress [37]. If hyperglycemia continues for a certain period in the early stages of diabetes, it may be difficult to suppress the subsequent progression of complications even if hyperglycemia is corrected by subsequent treatment [38]. Concepts of “metabolic memory” include many factors such as mitochondrial DNA damage, activation of protein kinase C, polyol pathway, increased production of advanced glycation end products (AGEs), overexpression of AGE receptors, increased anion superoxide formation, glycation of mitochondrial proteins, and hexosamine influx alterations [39]. Kowluru et al. compared metabolites in the blood and urine of rats in which glycemic control was started immediately after the onset of diabetes and in a group in which control was delayed for 6 months. They observed that oxidative stressors such as lipid peroxides (LPO), 8-hydroxy-2-deoxyguanosine (8-OhdG), glutathione (GSH), and urinary nitric oxide (NO) increased in the control delay group, but these substances in the rapid control group were no different from those in healthy rats [40]. Ceriello et al. observed increased vascular endothelial dysfunction, inflammation, and oxidative stress in patients with high HbA1c levels, but these nearly normalized in patients with low HbA1c levels who normalized blood glucose [41]. These reports suggest that the early initiation of good glycemic control may reduce oxidative stress and associated damage in the body, but the damage may progress if treatment is delayed. Quagliaro et al. found that intermittent hyperglycemia increased nitrotyrosine and 8-hydroxydeoxyguanosine, a marker of oxidative stress, and observed increased apoptosis in human venous endothelial cells, compared to stable high glucose condition [42]. Horvath et al. [43] observed that excessive glycemic changes stimulated nitrotyrosine production and caused endothelial dysfunction in streptozotocin-induced diabetic rats. Alterations in the retinal nerve fiber layer (RNFL) have also been reported in DR. Wang et al. [44] examined peripapillary vessel density and RNFL thickness using swept-source optical coherence tomography angiography (SS-OCTA) imaging in patients with nonproliferative DR, and they observed that peripapillary capillary vessel density and RNFL thickness were significantly lower in the superior temporal quadrants compared with healthy. Moreover, Cao et al. [45] observed that peripapillary capillary vessel density and RNFL thickness were significantly lower also in diabetic patients without DR compared to normal controls using OCTA.

Higher HbA1c levels, longer duration of diabetes, and use of insulin therapy for treatment have been reported to be risk factors for DR, and these may reflect poor glycemic control. Wat et al. reported that hypertension was positively correlated with the prevalence and progression of DR [46]. However, obesity, hyperlipidemia, sex, and smoking have not been established with clear associations with DR, as different studies have reported different results [46].

### 4.2. Pathophysiology of DN

The pathophysiologic mechanisms that lead to DN are multifactorial [19]. In the early stages of DN, hyperglycemia causes the release of vasoactive mediators such as VEGF, NO, glucagon, insulin-like growth factor 1 (IGF-1), and prostaglandins [47,48,49,50]. This causes dilation of the arterioles, leading to a temporary increase in the glomerular filtration rate. On the other hand, hyperglycemia causes the increased production of reactive oxygen species (ROS), activation of protein kinase C (PKC), and increased advanced glycation end products (AGEs), leading to vascular endothelial cell damage [50]. Damage to glomerular blood vessels results in increased glomerular permeability to macromolecules, leading to proteinuria. The transformation of IGF-1 also causes the hypertrophy of renal cells and the accumulation of extracellular matrix [51]. Sustained hyperglycemia also activates the renin-angiotensin system to form angiotensin II, which is involved in the development of DN and concomitant hypertension [52].

In patients with type 1 diabetes and type 2 diabetes, high HbA1c levels, longer duration of diabetes, and systemic hypertension are associated with an increased risk of development and progression of DN [53,54], similar to DR. In addition, dyslipidemia, obesity, age, smoking, and genetic factors are also risk factors for the development and progression of DN [53,54,55]. In a previous study, lower low-density lipoprotein cholesterol (LDL-C) and triglyceride (TG) levels were associated with a reduced risk of progression from moderate to severe albuminuria or end-stage renal disease [56]. The onset and severity of DN take some time. Microalbuminuria occurs 5 to 10 years after the onset of diabetes, followed by macroalbuminuria in 20 years and end-stage renal disease in 30 years [57].

Many studies have reported a significant association between DR and DN [58,59,60], and Kotlarsky et al. [60] observed that DN precedes DR. High HbA1c levels, longer diabetes duration, and systemic hypertension are considered common risk factors for DR and DN. Therefore, it is predicted that DR and DN are likely to develop and progress in parallel in patients with these factors.

### 4.3. Pathophysiology of DME

In the present study, we observed DME in half of the patients with DR but not in patients without DR. On the other hand, patients who develop DR often already have DN, because DN tends to precede DR [60].

Several reports have investigated the relationship between DME and DN. Acan et al. [61] reported that DN is a risk factor for developing DME. Koo et al. [62] reported that the serous type of DME was associated with albuminuria. However, there have been several reports that DN did not significantly affect DME development or progression [18,63]. On the other hand, there have been reports that dialysis improves DME. Takamura et al. [64] observed that the central retinal thickness (CRT) values were decreased after hemodialysis initiation without ocular treatments for DME in most eyes. Theodossiadis et al. [65] reported that average macular thickness and total macular volume were also reduced after hemodialysis in patients with DME. The mechanism by which hemodialysis promotes subretinal fluid absorption is currently unknown, but it is believed that the mutual flow of the excess fluid between the retina and the choroidal tissue or the retinal pigment epithelium may have improved after hemodialysis [64]. On the other hand, it has been proposed that SGLT-2 inhibitors may be effective in treating DME. Tatsumi et al. [66] observed that the administration of SGLT-2 inhibitors significantly reduced CRT in patients with untreated DME. SGLT2 is thought to be present in retinal pericytes and mesangial cells, and it has been reported that SGLT2 inhibitors may protect the retina by acting directly on retinal pericytes [67]. Therefore, rather than a diuretic effect, this direct action on retinal pericytes may contribute to the reduction of macular edema.

Several theories have been proposed regarding the mechanisms through which DME develops. They include retinal pigment epithelial (RPE) pump impairment, oxygen tension, Starling’s law, and BRB impairment. In areas with DME, microaneurysms and abnormal deep capillary networks are observed in the outer layer of the superficial nucleus, which is usually an avascular region [68]. The retinal pigment epithelium (RPE) has ion and fluid pumps for the reabsorption of subretinal fluid. This RPE pump is the sole extracellular fluid uptake mechanism in the fovea, and the dysfunction of this pump may be involved in the development of DME [69]. Decreased choroidal circulation may be the cause of RPE pump dysfunction [70]. An oxygen theory has also been proposed to explain the pathogenesis of DME [71]. Long-term hyperglycemia reduces retinal perfusion, resulting in a lower partial pressure of oxygen in the inner retina. Retinal arterioles dilate in an autoregulatory response, thus increasing the hydrostatic pressure within the capillaries and venules within the retina [72]. The resulting increase in intravascular pressure can damage capillaries [72]. In parallel, decreased retinal oxygen tension upregulates the synthesis of VEGF and other permeability factors and increases microvascular leakage.

VEGF is expressed not only by retinal cells but also by retinal pigment epithelium, glial cells, vascular endothelial cells, and pericytes under hypoxic conditions [73]. VEGF also causes endothelial hyperpermeability and endothelial cell proliferation [32]. VEGF is thought to play an important role in macular edema [74]. Funatsu et al. assessed fluorescence leakage in patients with DME and observed that VEGF levels in the vitreous humor were also elevated in patients with hyperfluorescence [75]. Therefore, the intravitreal injection of anti-VEGF agents (aflibercept and ranibizumab) has become the main treatment for DME, replacing conventional laser photocoagulation, vitrectomy, and steroid triamcinolone acetonide [76].

Starling’s law, which relates hydrostatic and oncotic pressures opposing each other, has long been advocated. It is speculated that this law causes elevated intravascular pressure and increased vascular permeability, resulting in the flow of water, ions, and macromolecules from the intravascular space into the extravascular space [20]. BRB is composed of retinal vasculature and retinal pigment epithelium (RPE). Endothelial cells are responsible for maintaining the BRB, and damage to these cells increases vascular permeability [77].

We observed that the prevalence of DME increased as the DR grade progressed in this study (A2: 32.7%, B1: 58.8%, B2: 71.4%, B4: 87.0%, B5: 100%). DME is a frequent manifestation of DR and can occur during any stage of the disease [77]. However, de Faria et al. [24] reported that it occurs more frequently as the duration of diabetes and severity of DR increase (early DR: 0%, nonproliferative DR: 64%, preproliferative DR: 79%, proliferative DR: 84%). As DR progresses, damage to capillaries and BRB impairment increase in the macula, and it is thought that the prevalence of DME also increases.

We consulted previous systemic reviews to examine DR prevalence in patients with type 2 diabetes in a general cohort [78,79,80,81,82,83] (Table 11). The global DR prevalence was 22.2–34.6% [78,79,80], but those of China and India were lower [81,82]. The DR prevalence in this study was high, presumably because it included patients who were referred for the purpose of assessment and treatment of DR.

### 4.4. Limitations

The blood pressure of each participant was measured at our institution but may not accurately reflect the usual blood pressure of each patient due to effects such as white-coat hypertension. Fundus findings in grades A1 and A2 and DME are reversible. Even if these findings had been temporarily present before, they may have disappeared at the time of examination. Diabetes onset time was estimated based on patient interviews and past medical records, but the duration of disease in each patient may have been longer because diabetes in its early stages is often asymptomatic.

Many patients with early-stage DR do not experience vision loss. Patients with diabetes require ophthalmology consultations to manage DR in parallel with medical treatment. However, many patients with diabetes only visit internal medicine, and the onset and progression of DR may be overlooked in such patients. The present study and previous studies [54] suggest a parallel progression of DR and DN. DN deterioration may be a predictor of DR onset and progression.

## 5. Conclusions

We observed that DR development involves the average HbA1c level, highest HbA1c level, and DN severity in addition to the duration of diabetes. DR severity correlated with the duration of diabetes, average HbA1c level, and DN severity. DN severity may be associated with DME development. Furthermore, DN progress may be a predictive indicator of DR development. Both DR and DN are microvascular complications of diabetes, and good long-term glycemic control may suppress the progression of both.

## Figures and Tables

**Figure 1 biomedicines-11-01502-f001:**
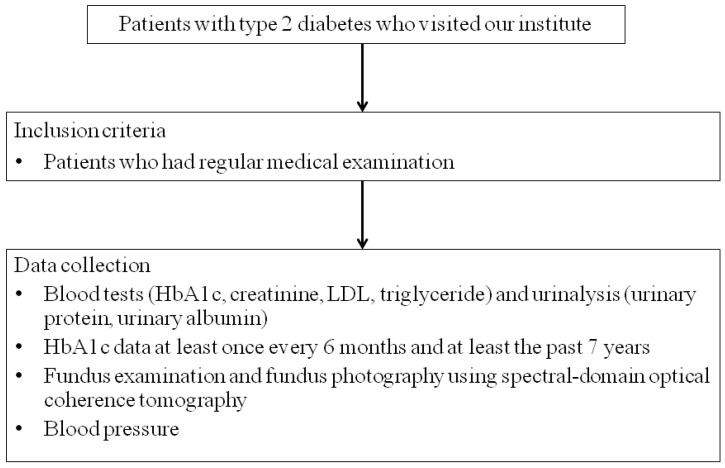
Flow chart of patient enrollment.

**Table 1 biomedicines-11-01502-t001:** Demographic data of patients with type 2 diabetes.

	With DR	Without DR	*p* Value
Male:Female	70:57	73:61	0.9
Average year	68.9 ± 10.8	71.2 ± 9.4	0.8
Diabetes duration	12.0 ± 2.7	9.8 ± 2.0	<0.0000001
DME	64 (50.4%)	0 (0%)	<0.0000001
Body mass index	24.2 ± 4.8	24.0 ± 4.5	0.9
Hypertension	61 (48.0%)	51 (38.1%)	0.1
Average HbA1c	7.9 ± 1.3	6.9 ± 0.7	<0.0000001
Highest HbA1c	9.5 ± 2.0	7.7 ± 1.3	<0.0000001
Serum LDL	117.3 ± 34.8	115.9 ± 31.7	0.4
Serum triglyceride	151.6 ± 137.3	139.8 ± 69.3	0.4
eGFR	56.2 ± 26.4	67.1 ± 17.0	0.00008
DN stage	2.4 ± 1.2	1.4 ± 0.6	<0.0000001
Ischemic heart disease	9 (7.1%)	5 (3.7%)	0.2
RAS inhibitors	22 (17.3%)	38 (28.4%)	0.04
SGLT-2 inhibitors	20 (15.7%)	23 (17.2%)	0.8

DR: diabetic retinopathy, DME: diabetic macular edema, DN: diabetic nephropathy, HbA1c: glycated hemoglobin, LDL: low-density lipoprotein, eGFR: estimated glomerular filtration rate, RAS: rennin-angiotensin-aldosterone system, SGLT: sodium/glucose cotransporter.

**Table 2 biomedicines-11-01502-t002:** Multivariate logistic regression analyses related to diabetic retinopathy development.

Factor	Odds Ratio	95% CI	*p* Value
Gender (male = 1, female = 0)	2.77	0.43, 17.9	0.3
Diabetes duration	1.65	1.14, 2.39	0.009
Body mass index	1.08	0.87, 1.32	0.5
Hypertension	0.42	0.09, 2.03	0.3
Average HbA1c	5.60	1.36, 23.1	0.02
Highest HbA1c	2.46	1.12, 5.38	0.02
Serum LDL	1.02	0.99, 1.04	0.2
Serum triglyceride	0.99	0.98, 1.00	0.07
DN stage	7.62	2.63, 22.1	0.0002
History of ischemic heart disease	0.55	0.03, 8.82	0.7
RAS inhibitors prescription	1.27	0.26, 6.18	0.8
SGLT-2 inhibitors prescription	1.57	0.30, 8.15	0.6

DN: diabetic nephropathy, HbA1c: glycated hemoglobin, LDL: low-density lipoprotein, RAS: rennin-angiotensin-aldosterone system, SGLT: sodium/glucose cotransporter.

**Table 3 biomedicines-11-01502-t003:** Univariate analysis with diabetic retinopathy development.

Factor	Odds Ratio	95% CI	*p* Value
eGFR	0.98	0.97, 0.99	0.0001
Albuminuria	5.23	3.14, 8.71	<0.00001
History of ischemic heart disease	0.55	0.03, 8.82	0.7
RAS inhibitors prescription	0.77	0.42, 1.40	0.4
SGLT-2 inhibitors prescription	1.55	0.79, 3.03	0.2

eGFR: estimated glomerular filtration rate RAS: rennin-angiotensin-aldosterone system, SGLT: sodium/glucose cotransporter.

**Table 4 biomedicines-11-01502-t004:** Multivariate regression analyses related to severity of diabetic retinopathy.

Factor	*t*-Statistic	95% CI	*p* Value
Gender (male = 1, female = 0)	–1.35	–0.85, 0.16	0.4
Diabetes duration	4.97	0.15, 0.36	<0.00001
Body mass index	–0.32	–0.08, 0.05	0.8
Hypertension	–1.99	–1.01, 0.005	0.06
Average HbA1c	3.25	0.14, 1.02	0.002
Highest HbA1c	–0.87	–0.43, 0.17	0.4
Serum LDL	1.60	–0.002, 0.02	0.1
Serum triglyceride	–0.26	–0.002, 0.002	0.8
DN stage	4.32	0.33, 0.89	0.00004
History of ischemic heart disease	1.30	–0.35, 1.68	0.2
RAS inhibitors prescription	0.71	–0.39, 0.82	0.5
SGLT-2 inhibitors prescription	0.66	–0.45, 0.89	0.5

DN: diabetic nephropathy, HbA1c: glycated hemoglobin, LDL: low-density lipoprotein. RAS: rennin-angiotensin-aldosterone system, SGLT: sodium/glucose cotransporter.

**Table 5 biomedicines-11-01502-t005:** Univariate analysis with severity of diabetic retinopathy.

Factor	*t*-Statistic	95% CI	*p* Value
eGFR	–4.96	–0.04, –0.02	<0.00001
Albuminuria	0.19	1.24, 1.99	<0.00001
History of ischemic heart disease	1.67	–0.16, 1.91	0.1
RAS inhibitors prescription	0.27	–0.68, 0.39	0.6
SGLT-2 inhibitors prescription	1.55	0.79, 3.03	0.5

eGFR: estimated glomerular filtration rate, RAS: rennin-angiotensin-aldosterone system, SGLT: sodium/glucose cotransporter.

**Table 6 biomedicines-11-01502-t006:** Multivariate logistic regression analyses related to diabetic macular edema development.

Factor	Odds Ratio	95% CI	*p* Value
Gender (male = 1, female = 0)	0.63	0.15, 2.61	0.5
Duration of diabetes	1.33	1.01, 1.75	0.04
Body mass index	0.95	0.79, 1.15	0.6
Hypertension	0.52	0.13, 2.04	0.3
Average HbA1c	5.52	1.27, 24.1	0.02
Highest HbA1c	0.75	0.35, 1.64	0.5
Serum LDL	1.01	0.93, 1.03	0.6
Serum triglyceride	1.00	0.99, 1.00	0.1
DN stage	2.80	1.37, 5.72	0.005
History of ischemic heart disease	2.63	0.27, 25.6	0.4
RAS inhibitors prescription	1.70	–0.27, 0.35	0.2
SGLT-2 inhibitors prescription	–0.12	–0.47, 0.22	0.2

DN: diabetic nephropathy, HbA1c: glycated hemoglobin, LDL: low-density lipoprotein. RAS: rennin-angiotensin-aldosterone system, SGLT: sodium/glucose cotransporter.

**Table 7 biomedicines-11-01502-t007:** Univariate analysis with diabetic macular edema development.

Factor	Odds Ratio	95% CI	*p* Value
eGFR	0.98	0.97, 0.99	0.009
Albuminuria	4.05	2.30, 7.11	<0.00001
RAS inhibitors prescription	0.76	0.36, 1.59	0.1
SGLT-2 inhibitors prescription	1.54	0.70, 3.37	0.3

eGFR: estimated glomerular filtration rate, RAS: rennin-angiotensin-aldosterone system, SGLT: sodium/glucose cotransporter.

**Table 8 biomedicines-11-01502-t008:** Multivariate regression analyses related to severity of diabetic macular edema.

Factor	*t*-Statistic	95% CI	*p* Value
Gender (male = 1, female = 0)	–0.53	–0.37, 0.21	0.6
Diabetes duration	2.57	0.02, 0.13	0.001
Body mass index	–0.21	–0.04, 0.03	0.8
Hypertension	–1.07	–0.44, 0.13	0.3
Average HbA1c	2.54	0.08, 0.68	0.01
Highest HbA1c	–0.64	–0.22, 0.11	0.5
Serum LDL	0.95	–0.005, 0.005	0.07
Serum triglyceride	–1.92	–0.002, 0.0004	0.06
DN stage	2.48	0.04, 0.36	0.002
History of ischemic heart disease	–0.08	–0.56, 0.52	0.9
RAS inhibitors prescription	0.25	–0.27, 0.35	0.8
SGLT-2 inhibitors prescription	–0.70	–0.47, 0.22	0.5

DN: diabetic nephropathy, HbA1c: glycated hemoglobin, LDL: low-density lipoprotein. RAS: rennin-angiotensin-aldosterone system, SGLT: sodium/glucose cotransporter.

**Table 9 biomedicines-11-01502-t009:** Univariate analysis with severity of diabetic macular edema.

Factor	*t*-Statistic	95% CI	*p* Value
eGFR	–2.44	–0.009, –0.001	0.02
Albuminuria	5.36	0.26, 0.55	<0.00001
RAS inhibitors prescription	–0.86	–0.30, 0.12	0.4
SGLT-2 inhibitors prescription	0.30	–0.19, 0.26	0.8

eGFR: estimated glomerular filtration rate, RAS: rennin-angiotensin-aldosterone system, SGLT: sodium/glucose cotransporter.

**Table 10 biomedicines-11-01502-t010:** Patients with diabetic macular edema development by diabetic retinopathy grade.

Fukuda Classification	Patients (Male/Female)	DME (Male/Female)	Prevalence
A0	134 (73/61)	0 (0/0)	0%
A1	3 (3/0)	0 (0/0)	0%
A2	49 (26/23)	16 (8/8)	32.7%
B1	34 (21/13)	20 (10/10)	58.8%
B2	7 (4/3)	5 (2/3)	71.4%
B3	0 (0/0)	0 (0/0)	-
B4	23 (11/12)	20 (9/11)	87.0%
B5	3 (2/1)	3 (2/1)	100%

DME: diabetic macular edema.

**Table 11 biomedicines-11-01502-t011:** Report on DR prevalence in patients with type 2 diabetes.

Report	DR Prevalence	Region	Year
Teo et al., 59 studies	22.3%	Global	2020
Yau et al., 35 studies	34.6%	Global	1980–2008
Cheloni et al., 10 studies	34.6%	Global	2008–2018
Song et al., 31 studies	18.5%	China	1990–2017
Brar et al., 10 studies	16.1%	India	1990–2021
Heiran et al., 109 studies	31%	Eastern Mediterranean	2019–2020
Present study	48.7%	Japan	2019–2022

DR: diabetic retinopathy.

## Data Availability

All data used for analysis are presented in the tables in this article. Data will be shared after ethics approval if requested by other investigators for the purpose of replicating the results.

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
