# Peer review of "Relationship between Diabetic Nephropathy and Development of Diabetic Macular Edema in Addition to Diabetic Retinopathy"

_biomedicines, 2023, doi:10.3390/biomedicines11051502_

Round 1

Reviewer 1 Report

Well structured, valuable manuscript. Important message could be that the body remembers periods
of bad CH metabolism, this should also be discussed.
Another useful potentials in this manuscript:
1. comparison of standard photos and OCT result and give an OCT staging for DME
2. management of DME, how does DN influence the treatment effectiveness
L80, Methods: What was the type of this study? Was it a true prospective study or a retrospective
consecutive case series? How were the data collected? What were the exclusion criteria?
L88 Section2.2 Is this a description of standard daily care?
L121, L 143: How was the eye selected for study?
Linear regression, multivariable regression: Obviously, the observed data may not follow normal
distribution. Were the errors after modelling normal?
L169 The authors should provide anonymity of the data. The patientsdata should be stored publicly
available.
L 229 The first response for high blood sugar is controversial in the literature, the changes in
neuroretina are also very important, please discuss it.

Author Response

Dear reviewer 1

Thank for your detailed review and kind comments.

biomedicines-2368688, MS TITLE: Relationship between diabetic nephropathy and development of diabetic macular edema in addition to diabetic retinopathy

Yukihisa Suzuki, and Motohiro Kiyosawa

Comments of Reviewer

Well structured, valuable manuscript. Important message could be that the body remembers periods of bad CH metabolism, this should also be discussed.

Response: It has been reported that when hyperglycemia persists for a certain period or longer in the early stages of diabetes, it becomes difficult to control subsequent diabetic complications, and the hypothesis of "metabolic memory" has been proposed to explain this. We added description about this in discussion section. [L439-]

Another useful potentials in this manuscript:
1. comparison of standard photos and OCT result and give an OCT staging for DME

Response: We adopted OCT-based staging (mild, moderate, and severe) proposed by Hwang et al. (2022) as DME staging in this study. We performed multivariate regression analyzes with DME severity and each parameter, and we observed significant correlations between DME severity and diabetes duration, average HbA1c level, and DN stage. We added the descriptions in Materials and Methods section and Result section. [L118-, L346-]

  1. management of DME, how does DN influence the treatment effectiveness

Response: There are reports that DN is a risk factor for developing DME, but there have been reports that DN did not significantly affect DME development or progression. On the other hand, there have been reports that dialysis improves DME. We added these description in Discussion section. [L521-]

L80, Methods: What was the type of this study? Was it a true prospective study or a retrospective consecutive case series? How were the data collected? What were the exclusion criteria?

Response: This study is retrospective consecutive case series. Blood tests (HbA1c, creatinine, LDL, triglyceride) and urinalysis (urinary protein, urinary albumin) were taken from ophthalmological examinations within 2 months, if available, or newly performed. Exclusion criteria were the lack of HbA1c data for the past 7 years, and unable to perform urinalysis due to chronic renal failure. We added these descriptions in Materials and Methods section. [L89-, L99-, L106-]

L88 Section2.2 Is this a description of standard daily care?

Response: We describe the required laboratory tests and historical HbA1c intervals for this study.

L121, L 143: How was the eye selected for study?

Response: In the present study, we selected the eye with more severe DR or DME as the target eye for each patient. We have corrected the description. [L147-, L184]

Linear regression, multivariable regression: Obviously, the observed data may not follow normal distribution. Were the errors after modelling normal?

Response: We used the Kolmogorov-Smirnov test to examine whether DR stage, average HbA1c level, high HbA1c level, and eGFR are normally distributed. We found that DR grade (P<0.001), average HBA1c level (P=0.001), high HBA1c level (P=0.002) were not normally distributed, while eGFR was normally distributed. We added the descriptions in Materials and Method section and Results section. [L149-, L227-]

L169 The authors should provide anonymity of the data. The patients’data should be stored publicly available.

Response: We provide data in a non-personally identifiable form. We added the description. [L190-]

L 229 The first response for high blood sugar is controversial in the literature, the changes in neuroretina are also very important, please discuss it.

Response: It is known that persistent hyperglycemia in the early stages of diabetes increases the incidence of subsequent complications, and the hypothesis of "metabolic memory" has been suggested to explain it. On the other hand, Wang et al. observed that peripapillary capillary vessel density and the retinal nerve fiber layer (RNFL) thickness were significantly lower in patients with nonproliferative DR compared with healthy using OCTA. Moreover, peripapillary capillary vessel density and RNFL thickness were significantly lower also in diabetic patients without DR compared to normal controls using OCTA. From these observations, it is considered that the reduction in RNFL thickness occurs from the early stage of diabetes onset. We added these descriptions in Discussion section. [L445-, L470-]

Reviewer 2 Report

The study is interesting, but there are some issues that need to be clarified:

1.                     Specify in the methods, if the data were collected prospectively or retrospectively.

2.                     Please provide a flow diagram of patient enrollment.

3.                     The analysis is possibly affected by residual confounding. Some important covariates are missing, such as history of cardiovascular disease, the use of RAS-inhibitors, the use of SGLT-2 inhibitors, etch. Please try to make your multi-variate models stronger. The univariate analyses need to be presented as well.

4.                     The association between diabetic nephropathy and diabetic retinopathy needs to be explored more closely. Please explore the independent association of DR with albuminuria and eGFR.

5.                     This study excluded patients with type 1 diabetes – justify your decision.

6.                     The study included patients referred to your department for assessment of DR. The estimated prevalence rate of DR in this cohort may not accurately reflect the actual prevalence of DR in the general diabetic population. This is a clear limitation. Provide comparison with other studies in a separate table in the appendix.

Author Response

Dear reviewer 2

Thank for your detailed review and kind comments.

biomedicines-2368688, MS TITLE: Relationship between diabetic nephropathy and development of diabetic macular edema in addition to diabetic retinopathy

Yukihisa Suzuki, and Motohiro Kiyosawa

Comments of Reviewer

Specify in the methods, if the data were collected prospectively or retrospectively.

Response: Data were collected retrospectively. We added the description in Method section. [L89-]

Please provide a flow diagram of patient enrollment.

Response: We provided a flow diagram of patient enrollment. [L135-, Table1]

The analysis is possibly affected by residual confounding. Some important covariates are missing, such as history of cardiovascular disease, the use of RAS-inhibitors, the use of SGLT-2 inhibitors, etch. Please try to make your multi-variate models stronger. The univariate analyses need to be presented as well.

Response: We performed Multivariable logistic regression analyzes between a diagnosis of type 2 diabetes and the parameters and regression analysis between DR stage and the parameters, including history of cardiovascular disease, use of RAS-inhibitors, and use of SGLT-2 inhibitors. [L128-, L232-]

The association between diabetic nephropathy and diabetic retinopathy needs to be explored more closely. Please explore the independent association of DR with albuminuria and eGFR.

Response: Urinary albumin levels were classified into A1 (less than 30 mg/day), A2 (30-299 mg/day), and A3 (300 mg/day or more) according to the KINGO CKD guideline 2012. We examined independent association of DR with albuminuria and eGFR. [L126-, L236-, L275-, L312-, L350-]

This study excluded patients with type 1 diabetes – justify your decision.

Response: Type 1 diabetes and type 2 diabetes are inherently different pathologies. It has been also reported that the incidence of DR differs between type 1 diabetes and type 2 diabetes (Li 2021; Matuszewski 2020). Therefore, we focused only type 2 diabetes in this study. We added the description in Method section. [L91-]

The study included patients referred to your department for assessment of DR. The estimated prevalence rate of DR in this cohort may not accurately reflect the actual prevalence of DR in the general diabetic population. This is a clear limitation. Provide comparison with other studies in a separate table in the appendix.

Response: According to other systemic reviews, DR prevalence in patients with type 2 diabetes ranged from 16.1% to 34.6%, lower than 48.7% in the present study. This is presumed to be due to the inclusion of patients referred for the purpose of assessment and treatment of DR at our institution. [L581-, Table 7]

Round 2

Reviewer 2 Report

The revisions are satisfactory and the overall merit of the paper has been improved.